# En Bloc Bipolar Prostate Enucleation Using the Mushroom Technique with Early Apical Release: Short-Term Outcomes

**DOI:** 10.3390/medicina61101859

**Published:** 2025-10-16

**Authors:** Zoltán Kiss, Mihály Murányi, Alexandra Barkóczi, Gyula Drabik, Attila Nagy, Tibor Flaskó

**Affiliations:** 1Department of Urology, University of Debrecen, 4032 Debrecen, Hungary; muranyi.mihaly@med.unideb.hu (M.M.); dr.barkoczi.alexandra@med.unideb.hu (A.B.); drabik.gyula@med.unideb.hu (G.D.); flash@med.unideb.hu (T.F.); 2Department of Health Informatics, Faculty of Health Sciences, University of Debrecen, 4032 Debrecen, Hungary; nagy.attila@sph.unideb.hu

**Keywords:** anatomical endoscopic enucleation of the prostate, benign prostatic hyperplasia, bipolar prostate enucleation, mushroom technique

## Abstract

*Background and Objectives:* While transurethral resection of the prostate remains the gold standard for surgical treatment of benign prostatic hyperplasia, anatomical endoscopic enucleation of the prostate provides a safe, durable, and size-independent alternative. Our study introduces a specific technical innovation, i.e., en bloc bipolar prostate enucleation performed exclusively via sheath-tip mechanical dissection without the use of a dedicated enucleation loop, combined with the mushroom technique and early apical release. *Materials and Methods:* Between January 2018 and May 2023, 252 patients with prostate volumes > 30 mL and significant lower urinary tract symptoms underwent en bloc bipolar prostate enucleation via the mushroom technique with early apical release. Data were retrospectively evaluated to assess perioperative results, postoperative outcomes, and complications. *Results:* The median age of the cohort was 70 (65–76) years, with a median prostate volume of 60 (40–88.5) mL. The median operative time was 40 (30–70) min, and the median weight of enucleated tissue was 34 (16.5–60) g. Significant improvements were observed in the International Prostate Symptom score, Quality of Life score, maximum flow rate, average flow rate, and postvoid residual urine at 12 months (*p* < 0.001). The rate of transient stress urinary incontinence decreased from 19.44% at 1 month to 2.38% at 12 months. *Conclusions:* En bloc bipolar prostate enucleation using the mushroom technique is a safe and effective treatment for benign prostatic hyperplasia, yielding significant improvements in urinary symptoms and flow rates, with a manageable complication profile. Further multicenter studies are needed to confirm these findings.

## 1. Introduction

Various minimally invasive techniques have been developed for the surgical treatment of benign prostatic hyperplasia (BPH); however, transurethral resection of the prostate (TURP) remains the gold standard. Despite its high morbidity, TURP has demonstrated enduring clinical relevance over time [1]. Recent advancements in surgical techniques—such as anatomical endoscopic enucleation of the prostate (AEEP)—have emerged as promising alternatives, offering advantages such as reduced complication rates and improved outcomes. AEEP is a safe, durable, and size-independent method that adopts the principles of open prostatectomy in a transurethral setting [2,3,4,5]. Tissue removal typically involves the use of morcellators; however, these tools are relatively expensive and may not be available in every department. Additionally, morcellation has a steep learning curve and potential complications.

The mushroom technique serves as an alternative method for removing the adenoma [6]. In this technique, the adenoma is not released into the bladder; instead, a narrow, mushroom-like pedicle is preserved at the bladder neck, keeping the avascularized lobes in place. The lobes can then be resected quickly and easily, allowing the pieces to be washed out.

In this study, we aimed to evaluate the efficacy and safety of our novel technique, characterized by en bloc bipolar prostate enucleation performed exclusively via sheath-tip mechanical dissection without the use of a dedicated enucleation loop. Early apical release is used to preserve the sphincter mucosa, and the prostatic tissue is removed using the mushroom technique.

To our knowledge, this specific combination of en bloc enucleation, sheath-tip-only dissection, and the mushroom technique has not been previously described. We hypothesize that this technique will demonstrate comparable efficacy and safety to those that utilize morcellators, with significant improvements in urinary symptoms and a manageable complication profile.

## 2. Materials and Methods

Between January 2018 and May 2023, 252 patients underwent en bloc bipolar prostate enucleation using the mushroom technique with early apical release at a single tertiary university hospital. Following approval from the Regional Institutional Research Ethics Committee (IRB No. DERKEB/IKEB 6977-2024) and the acquisition of written informed consent, data were retrospectively collected. All procedures were performed by a single surgeon experienced in endourology. The inclusion criteria were as follows: prostate volume > 30 mL, maximum flow rate (Qmax) < 15 mL/s, postvoid residual (PVR) urine > 100 mL, International Prostate Symptom score (IPSS) > 7, urinary retention, and failure of medical therapy. Patients who opted not to undergo medical therapy were also included.

Patients diagnosed with neurogenic bladder, urethral stricture, or prostate cancer, as well as those who had previous urethral surgery, were excluded. Patient characteristics—including body mass index (BMI), American Society of Anesthesiologists (ASA) score, and prostate-specific antigen (PSA) levels—were collected from clinical records. All PSA measurements were performed in the same institutional laboratory using a standardized immunoassay platform, thereby ensuring assay uniformity throughout the study period. Prostate volume was measured via transabdominal ultrasound (Consona N9 Diagnostic Ultrasound System; Mindray, Shenzhen, China) using the ellipsoid formula. Each patient was assessed using the IPSS and Quality of Life (QoL) questionnaires, uroflowmetry, and PVR measurements. Uroflowmetry was performed using the Urodoc device (Urodoc, Budapest, Hungary), with one measurement obtained in each case. PVR urine was assessed by transabdominal ultrasound (Consona N9 Diagnostic Ultrasound System; Mindray, Shenzhen, China) using the ellipsoid formula. All measurements were performed by the same physician to ensure standardization. Erectile function was evaluated using the International Index of Erectile Function (IIEF-5) questionnaire. Urine cultures and laboratory parameters (natrium, creatinine, hemoglobin) were also analyzed.

### 2.1. Surgical Technique

Intravenous antibiotics were administered to the patients 30 min before the start of surgery. Typically, a broad-spectrum third-generation cephalosporin, such as ceftriaxone (2 g), was used. In cases of positive preoperative urine culture, targeted parenteral antibiotic therapy (most commonly a cephalosporin) was initiated the day before surgery and continued during hospitalization.

The procedure was performed under spinal anesthesia with the patient in the lithotomy position. All interventions were carried out using the following instruments from Olympus Winter & IBE GmbH, Hamburg, Germany:ESG-400 high-frequency electrosurgical generator with settings of 200 W for cutting and 120 W for coagulation, effect: 2Outer sheath: 27-Fr, rotatable, continuous flow, with two stopcocksResection sheath: 24-FrOptic: 12°, 4 mm, HDHigh-frequency bipolar electrode loop (PlasmaLoop Medium)Plasma-OvalButton electrode: in cases of severe intraoperative bleedingIrrigation fluid: preheated saline, irrigation height: 60 cm above the patient.

First, an accurate cystoscopy was performed to identify the ureteral orifices. Next, we proceeded to the prostatic urethra, where an omega-shaped incision was made from the verumontanum to the 12 o’clock position to release the sphincter mucosa. Thereafter, the space between the capsule and the adenoma was opened with the tip of the sheath, starting laterally from the sides of the verumontanum. A key technical point is that the tip of the sheath must be placed between the verumontanum and adenoma, and then gently pushed laterally and slightly downwards. This maneuver reveals the smooth, vascularized layer that confirms entry into the correct plane. It should be emphasized that the tip of the sheath is not directly visible during the procedure; instead, it lies at the edge of the endoscopic screen. Therefore, it is essential to advance close to the capsule with the tip of the sheath in order to remain in the correct plane throughout the enucleation.

After identifying the correct plane, the adenoma was detached from the capsule via blunt mechanical detachment, resulting in complete enucleation of the left lobe.

When the vertical fibers of the bladder neck became visible, the sheath was advanced to enter the bladder at the 2 o’clock position. The adenoma was enucleated from the 2–6 o’clock position, with all bleeding coagulated on the capsule using the loop. Bleeding is most often encountered from perforating arteries between the 2–5 o’clock and 7–10 o’clock positions, particularly in larger glands. All visible bleeders were carefully coagulated before proceeding, as even minor persistent oozing may impair visualization. Compared with resection, enucleation usually transects fewer vessels; still, meticulous hemostasis remains critical, especially in prolonged procedures.

When a median lobe was present, it was enucleated at the 6 o’clock position, with particular attention paid to the ureteric orifices.

The procedure continued with the right lobe; the bladder was entered at the 10 o’clock position, and the adenoma was enucleated from above down to the 6 o’clock region. At this point, the two enucleation surfaces were connected. A 1–2 cm mushroom-like pedicle was preserved at the 6 o’clock position on the bladder neck to secure the adenoma in the prostatic fossa. The apical part of the adenoma was then enucleated between the 10 and 2 o’clock positions.

The avascularized adenoma was cut at high speed using the loop, and the adenoma slices were washed out. Final coagulation of the capsule was performed with special attention to the bladder neck, and the preservation of the sphincteric mucosa was verified.

In cases of severe intraoperative bleeding, the OvalButton electrode was used to achieve proper hemostasis. A three-way catheter was then inserted, and continuous bladder irrigation was initiated. The surgical technique is illustrated in Figure 1.

### 2.2. Outcome Measures

The operative time and weight of the enucleated prostatic tissue were measured and recorded, from which the enucleation efficacy (g/min) was calculated. On the first postoperative day, hemoglobin, natrium, and creatinine levels were monitored, and the duration of bladder irrigation, catheter dwell time, and length of hospital stay were documented. Complications were assessed according to the Clavien–Dindo classification.

### 2.3. Follow-Up

Patients with a minimum follow-up duration of 12 months were included in the study. Follow-up was conducted at 1, 3, 6, and 12 months, during which PSA levels, uroflowmetry, PVR, IPSS, QoL, and IIEF-5 scores were measured. The presence of transient stress urinary incontinence (TSUI)—defined as the need for pad use—was also documented.

### 2.4. Statistical Analysis

The normality of continuous variables was evaluated using the Shapiro–Wilk test. Categorical variables are described by proportions, while continuous variables are presented as medians and interquartile ranges. For repeated assessments of continuous outcomes (baseline, 1, 3, 6, and 12 months), we applied linear mixed-effects models with a random intercept for subjects and fixed effects for time. Effect sizes with 95% confidence intervals were reported from these models. In sensitivity analyses, the Friedman test was used as a non-parametric alternative, while post hoc pairwise comparisons between time points were explored using Wilcoxon signed-rank tests with multiplicity adjustment. Pearson’s Chi-squared test was used to explore associations between categorical variables. Statistical significance was defined as *p* < 0.05. All statistical analyses were performed using Intercooled Stata v18.0 (Stata Statistical Software: Release 18; StataCorp LLC, College Station, TX, USA).

## 3. Results

The median age of the patients was 70 (65–76) years, and the median BMI was 27.65 (24.7–30.15) kg/m^2^. The ASA scores were distributed as follows: ASA score 1, *n* = 6 (2%); ASA score 2, *n* = 134 (53%); ASA score 3, *n* = 108 (43%); and ASA score 4, *n* = 4 (2%). Antiplatelet therapy was administered to 44 (17%) patients, 165 (65%) patients were on preoperative medical therapy for BPH, and six (2%) patients had undergone previous TURP. Concomitant bladder stones were noted in eight (3%) patients, for whom transurethral cystolitholapaxy using an ultrasound lithotripter (ShockPulse-SE Lithotripsy System; Cybersonics, Inc., Erie, PA, USA) was also performed in the same session. The median prostate volume was 60 (40–88.5) mL. Urinary retention was detected in 133 (52%) patients, and the median preoperative PSA was 4.17 (1.79–7.81) ng/mL. The median IPSS, QoL, Qmax, average flow rate (Qave) and PVR urine were 21 (16–25), 6 (5–6), 8.4 (5.9–11.75) mL/s, 3.7 (2.8–5.7) mL/s, and 120 (57.5–250) mL, respectively. The preoperative urine culture positivity rate was 50% (125 patients). The median IIEF-5 score was 12 (5–18). The median preoperative natrium, creatinine, and Hgb levels were 140 (139–141) mmol/L, 82 (72–95) µmol/L, and 146 (138–153) g/L, respectively. Baseline characteristics according to prostate volume subgroups and urinary retention status are presented in Table 1 and Table 2.

The median operative time was 40 (30–70) min, and the median weight of the enucleated prostate tissue was 34 (16.5–60) g. The overall enucleation efficacy was 0.76 (0.6–1) g/min. A significant difference in enucleation efficacy was observed between prostate sizes < 80 mL and >80 mL (0.66 [0.53–0.86] g/min vs. 1 [0.76–1.11] g/min [*p* < 0.001]), indicating that larger prostate sizes resulted in more effective enucleation. The median bladder irrigation time was 24 (20–24) h. The mean hemoglobin drop was 13.28 g/L (*p* < 0.001). One patient (0.4%) required a blood transfusion, which was anticipated due to preoperative anemia. Changes in median natrium (140 [139–141] mmol/L vs. 139 [138–141] mmol/L, *p* = 0.011) and creatinine levels (82 [72–95] µmol/L vs. 79.5 [68–95] µmol/L, *p* = 0.002) were significant.

The median catheter dwell time was 3 (2–3) days, and catheter removal was successful in all patients. The median length of hospital stay was 4 (3–4) days. According to histology reports, 233 (92%) patients were diagnosed with BPH, while 19 patients (8%) had incidental prostate cancer (iPCa). Among cases of iPCa, 16 (84%) were classified as low-risk (Gleason score 3 + 3 = 6), whereas 2 (11%) and 1 (5%) case had a Gleason score of 3 + 4 = 7 and 4 + 5 = 9 adenocarcinoma, respectively. Active surveillance was implemented in the low-risk cases, while the other three patients were managed with radiotherapy and androgen deprivation therapy.

Regarding Clavien–Dindo grade I complications, gross hematuria was observed in 8% (21/252) managed with prolonged bladder irrigation, catheter traction, and tranexamic acid. Capsule perforation occurred in 5% (13/252) managed with prolonged catheterization. Regarding Clavien–Dindo grade II complications, fever was noted in 3% (8/252) treated with parenteral antibiotic therapy, and two (2/252, 1%) patients treated with parenteral antibiotics and NSAIDs required readmission due to epididymitis. All infectious complications occurred in long-term catheter carriers with positive preoperative urine cultures. Clavien–Dindo grade IIIb complications occurred in 3% (7/252) patients who required clot evacuation and coagulation, two (2/252, 1%) patients who required internal urethrotomy due to urethral stricture, and one (1/252, 0.4%) patient who required transurethral bladder neck resection due to bladder neck sclerosis. No Clavien–Dindo grade IIIa, IV, or V complications occurred.

The percentage reduction in PSA was calculated for each patient as (postoperative PSA—preoperative PSA)/preoperative PSA. These individual percentage changes were then averaged across the cohort.

During follow-up, voiding parameters improved significantly, including Qmax, Qave, PVR urine and IPSSs. However, a significant decline was observed in IIEF-5 at 12 months. In our cohort, 73 patients had a baseline IIEF-5 score ≥ 17 (21 [19–23]). At the 6-month follow-up, 41 of these patients were available for assessment, with a median IIEF-5 of 16 [8–22], indicating a decrease in erectile function among previously potent men.

Figure 2 presents a flowchart of patient attendance at each follow-up visit, while follow-up data are shown in Table 3 and Table 4, Figure 3.

## 4. Discussion

The first milestone of the evolution of TURP was laid down by Hiraoka and Akimoto in 1989 [7]. Later, the development of laser technology enabled Peter Gilling to perform the first holmium laser enucleation (HoLEP) in 1998, where transurethral tissue morcellation was applied to remove the adenoma from the bladder [8]. While morcellators have facilitated tissue removal, they are relatively expensive and not universally available, presenting a barrier to their widespread adoption.

The mushroom technique, introduced by Hochreiter in 2002 in HoLEP procedures, offers a viable alternative for adenoma removal [6]. By preserving a narrow, mushroom-like pedicle at the bladder neck, this technique prevents the adenoma from being released into the bladder, allowing for efficient resection and easy removal of the avascularized lobes.

The superiority of anatomical endoscopic prostate enucleation over conventional resection (TURP) and open prostatectomy has been demonstrated by numerous authors [9,10]. In cases of TURP, larger prostate volumes are associated with an increased risk of complications [11]. By contrast, enucleation is a size-independent procedure. Owing to its advantages, some authors now consider enucleation to be the new gold standard for the surgical treatment of BPH [12]. Various energy sources can be used, including bipolar, Ho:YAG, Tm:YAG, TFL, and GreenLight laser; however, the surgical technique itself is more critical to the success of the operation than the energy source used [13].

The three-lobe technique was initially employed, followed by the two-lobe technique; the prostate lobes were enucleated separately. Recently, the en bloc technique with early sphincter release has gained widespread acceptance. This approach offers several advantages, including improved visualization, quicker identification of the prostate capsule, shorter operation times, reduced energy consumption, and a faster learning curve. The en bloc technique makes it easier to preserve the integrity of the sphincter mucosa, and early sphincter release at the beginning of the surgery is particularly important for reducing the incidence of postoperative TSUI [14,15,16].

Transurethral bipolar prostate enucleation using the mushroom technique combines enucleation and resection in a single procedure. This method may offer cost advantages, as it eliminates the need for a morcellator. Additionally, our technique does not require a dedicated enucleation loop; instead, the entire enucleation process is performed using the tip of the resectoscope sheath, which could potentially reduce additional expenses. However, we acknowledge that this assumption is not supported by a formal cost analysis and should therefore be regarded as hypothetical. From an educational perspective, bipolar prostate enucleation may also represent a suitable training option, as most urologists are more familiar with bipolar than laser resectoscopes. Additionally, transitioning from enucleation to conventional resection may be more straightforward if complications arise during surgery.

Several studies have examined the combination of bipolar prostate enucleation with the mushroom technique [17,18,19]. In these approaches, enucleation begins with the resection of a 6 or 12 o’clock channel, or both. The lobes are divided through resection and then enucleated separately. Dedicated enucleation loops are used, which are swapped for a cutting loop during surgery to resect the adenoma. In these techniques, the loop may be exchanged up to four times during a single procedure. In our technique, the procedure is divided into two main parts, i.e., en bloc enucleation, followed by resection of the avascularized lobes. This technique does not involve resection between the lobes. The tip of the resectoscope sheath is employed for enucleation, eliminating the need to change loops during surgery. Resection of a channel at 6 or 12 o’clock has drawbacks, as irrigation fluid can escape into the bladder. Additionally, if a median lobe is present, it must also be resected to create a 6 o’clock channel, which can be time-consuming and may lead to bleeding. By contrast, the en bloc technique allows for irrigation in the enucleated space alone. Additionally, the median lobe is not resected at the beginning of the procedure; instead, it is enucleated en bloc and subsequently resected together with the lateral lobes after completion of the enucleation.

Regarding improvements in IPSS, QoL, Qmax, Qave and PVR urine, the results of our technique are comparable to those reported in the international literature [20]. Complete removal of the transitional zone during enucleation results in a 60–90% reduction in PSA levels [21]. Similarly, in our cohort, PSA decreased by 60% at 1 month and by 77% at 3, 6, and 12 months, which is consistent with the reported range.

Although enucleation efficacy (g/min) appeared higher in larger prostates, this finding should be interpreted with caution. The apparent increase may partly reflect a denominator effect, as the removal of larger tissue volumes does not always result in a proportionate increase in operative time. Consequently, the observed association could represent a mathematical artifact rather than a true improvement in surgical efficiency. Moreover, case-mix factors such as prostate size, the presence of a median lobe, and urinary retention status may all influence surgical dynamics, and these variables should be considered when comparing efficiency outcomes across subgroups.

One of the most common complications, particularly regarding the learning curve, is capsule perforation. The rate of capsule perforation observed in our study aligns with findings in the literature [22]. Capsule perforations occur more frequently in smaller prostates, as the capsule is typically less well defined, and dissection of the plane is more challenging compared with larger glands [23]. Liu et al. [24] found that prostate volume was the strongest risk factor for an indistinct plane. According to their study, a prostate volume < 54 mL was associated with the highest likelihood of indistinct plane formation during HoLEP. In our series, all perforations occurred in prostates smaller than 60 mL. Enucleation was completed in all cases, and during the resection, a suction tube was connected to the outflow of the resectoscope to reduce intravesical pressure, thereby minimizing the risk of hypervolemia.

Although rare, one of the most challenging complications to manage is bladder neck sclerosis, which occurs in approximately 0.8% of patients after AEEP [25]. Several risk factors for bladder neck sclerosis have been identified, including a smaller prostate volume (<30 mL), lower preoperative PSA value, smaller amount of resected tissue, reoperation required for macroscopic hematuria, prolonged postoperative catheterization, and positive preoperative urine culture [26,27]. Sun et al. [28] compared enucleation and resection techniques in small prostates and found that the incidence of bladder neck sclerosis was significantly lower following enucleation (1.8% vs. 13.6%, *p* = 0.045). They explained their findings by suggesting that enucleation more effectively preserves the anatomical integrity of the bladder neck and reduces the likelihood of thermal damage. In our study, a single instance of bladder neck sclerosis was observed in a patient who required recoagulation after surgery due to macroscopic hematuria.

In addition to bladder neck sclerosis, urethral stricture may also develop following enucleation; while this can affect any part of the urethra, it typically occurs at the level of the meatus. In our study, both cases of urethral stricture occurred in the bulbar urethra and were successfully managed with internal urethrotomy. The incidence of urethral stricture after HoLEP was reported to be 1.6% [25]. A meta-analysis by Zhang et al. [10] demonstrated that the incidence of urethral stricture is significantly lower after AEEP compared with TURP. Preoperative urethral dilation, repeated use of lubricants, application of rotatable resectoscopes, and shorter operative times may all contribute to reducing the risk of urethral stricture formation [29,30].

Intra- and postoperative bleeding may occur during AEEP; however, the extent of bleeding is generally lower compared with TURP or open prostatectomy, as the adenoma is precisely detached from the capsule while simultaneously performing coagulation [31]. Factors influencing the amount of blood loss include prolonged operative time, inadequate intraoperative coagulation, preexisting coagulopathy, and capsule perforation [32]. In cases of severe intraoperative bleeding, the use of a bipolar OvalButton electrode for coagulation is a viable option. In our practice, postoperative hematuria is initially managed conservatively with intravenous tranexamic acid administration, catheter traction, and prolonged bladder irrigation. If these measures prove insufficient, recoagulation—which was necessary in 2.8% of cases in our series—is required.

Erectile dysfunction (ED) is also a known complication after AEEP. According to the results of Enikeev et al. [33], erectile function remained stable in 56% of patients, deteriorated in 18%, and improved in 26% following Thulium Laser Enucleation of the Prostate (ThuLEP). The decline in erectile function can be attributed to thermal damage to the prostatic capsule and neurovascular bundles [34]. Elshal et al. [35] investigated the effect of low-energy (2 J/25 Hz) versus high-energy (2 J/50 Hz) HoLEP on sexual function; they found no significant differences, concluding that the applied energy does not substantially affect erectile function. Naturally, the fact that with the improvement of urinary symptoms after surgery, medications that negatively affect sexual function may be discontinued should not be overlooked. A multicenter study including 235 patients who underwent HoLEP showed that increasing age was independently associated with a higher likelihood of declining erectile function at the 12-month follow-up (*p* = 0.03). This observation suggests that age-related vulnerability plays a more important role than the surgical technique itself in the onset of postoperative ED. Furthermore, patients with an ASA score > 2 were more prone to erectile deterioration 1 year after surgery, underlining the impact of preoperative physical condition on functional outcomes [36]. At baseline, erectile function already indicated mild-to-moderate ED in our study population. At the 12-month follow-up, this significantly declined. Naturally, for the reasons detailed above, surgery may have contributed to the deterioration of erectile function; however, it should also be emphasized that our study involved an elderly patient cohort with multiple comorbidities, as 42.86% of the patients were classified as ASA grade 3. Therefore, providing appropriate and detailed patient counseling prior to surgery is of paramount importance. In this context, minimally invasive surgical therapies have emerged as attractive alternatives, particularly owing to their favorable profile regarding the preservation of sexual function [37]. However, their main limitation lies in their higher costs and higher rate of reoperations (compared with enucleation techniques), which raises concerns for both healthcare systems and patients, who may have additional physical and psychological burden [38]. Hence, while minimally invasive surgical therapies certainly play a role in the surgical management of BPH in carefully selected cases, durable relief of lower urinary tract symptoms remains the primary therapeutic goal.

A significant disadvantage of enucleation, regardless of the energy source, is the occurrence of early TSUI [39]. To mitigate this, an omega-shaped incision is made from the paracollicular region to the 12 o’clock position at the start of the surgery, reducing sphincter stretching and minimizing the risk of tearing the sphincteric mucosa. The rate of TSUI observed in our study is consistent with the rates reported in the literature [40,41]. In our study, the detection rate of iPCa was 7.54%, which is comparable to data in the international literature. Cheng et al. [42] reported an 8% incidence of prostate cancer following AEEP procedures; interestingly, this does not differ from the rate observed after TURP. Similarly, Herlemann et al. [43] found no significant difference in the incidence of iPCa between HoLEP and TURP. This can be explained by the fact that larger prostates are more likely to be diagnosed as BPH after the final histological analysis. Predictive factors for iPCa include advanced age, elevated preoperative PSA levels, smaller resected tissue volume, higher PSA density, and diabetes mellitus [44,45]. In line with findings from the literature, most iPCa cases identified in our study represented low-risk prostate cancer (Gleason score 3 + 3 = 6) [46]. The therapeutic strategy for iPCa should always be individually determined considering the clinical characteristics and patient preferences; however, as most cases represent low-risk disease, and the malignant component constitutes only a small proportion of the specimen, active surveillance is generally sufficient. Whether preoperative multiparametric magnetic resonance imaging (mpMRI) should be routinely performed in patients with elevated preoperative PSA levels to improve the detection and risk stratification of iPCa remains an interesting and clinically relevant question. Porreca et al. [47] investigated the relationship between a negative preoperative mpMRI and the incidence of iPCa after HoLEP. They found that the incidence of iPCa was significantly lower in patients with a negative preoperative mpMRI compared to those without mpMRI (6.2% vs. 14.8%, *p* = 0.03). Nevertheless, no significant differences were observed between the negative mpMRI and no-mpMRI groups regarding the proportion of low-risk Gleason score 3 + 3 = 6 tumors.

Based on our experience, some urologists hesitate to begin enucleation procedures because they lack access to a laser or morcellator. We hope that our technique will encourage them by demonstrating that there is an alternative method for enucleation that does not require a laser, morcellator, or even a dedicated enucleation loop.

The limitations of our study include its single-center, retrospective design, the absence of a comparator arm, and the relatively short follow-up period limited to 12 months.

Additionally, prostate volume was measured by transabdominal ultrasound using the ellipsoid formula, which is less accurate than transrectal ultrasound. This potential measurement bias may be more pronounced in certain prostate size ranges, and future studies including sensitivity analyses stratified by size bands are warranted. Nevertheless, the results of a recent multicenter study that directly compared HoLEP combined with morcellation and bipolar enucleation using the mushroom technique confirmed that both procedures were safe and effective, with comparable functional improvements at 12 months. HoLEP was associated with shorter operative times and hospital stays, while IPSS, PSA, and PVR outcomes were similar between the groups. Despite the single-center design of our study, these findings support the external validity of our present results [48].

Further research with larger, multicenter trials and longer follow-up periods is warranted to validate these findings and establish the long-term efficacy and safety of this approach.

## 5. Conclusions

In conclusion, our study demonstrates that en bloc bipolar prostate enucleation using the mushroom technique with early apical release is a safe, feasible and cost-effective surgical option for treating BPH in patients with significant lower urinary tract symptoms. The results indicate significant improvements in patient-reported outcomes, including reductions in IPSS and PVR urine, alongside increases in Qmax and Qave over a 12-month follow-up period. The incidence of TSUI and other complications is consistent with existing literature; however, these findings should be interpreted with caution as the present study only describes short-term outcomes. Longer follow-up is necessary to assess the functional durability of the technique and late complications, such as stricture rates. Future prospective or multicenter studies with extended observation periods are warranted to confirm the long-term safety, durability, and reproducibility of this surgical approach, as well as to further investigate the learning curve associated with its adoption.

## Figures and Tables

**Figure 1 medicina-61-01859-f001:**
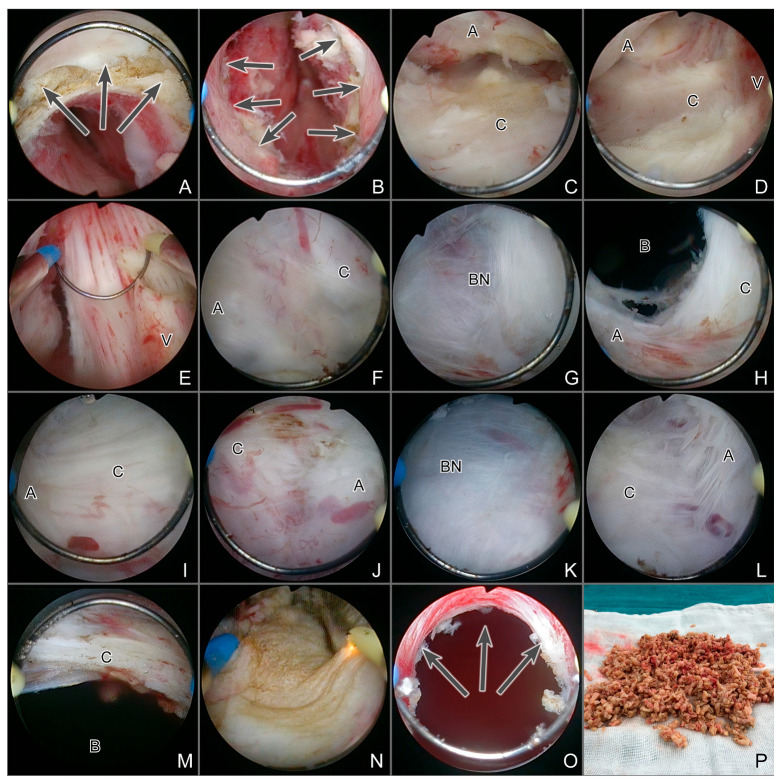
Steps involved in en bloc bipolar prostate enucleation using the mushroom technique with early apical release. (**A**) Mucosal incision at the 12 o’clock position for early apical release; The arrows in the image indicate the site of sphincter release at the 12 o’clock position; (**B**) Mucosal incision at the lateral lobes for early apical release; The arrows in the image indicate the site of sphincter release at the lateral lobes; (**C**) Opening the space between the capsule and the adenoma at the 5 o’clock position; (**D**) Opening the space between the capsule and the adenoma at the 7 o’clock position; (**E**) Connecting the opened spaces with the incision of the mucosa in front of the verumontanum; (**F**) Enucleation of the left lobe with the tip of the sheath; (**G**) Pathognomonic view before entering the bladder at the 10 o’clock position; (**H**) Enucleation of the left lobe from upside down; (**I**) Enucleation at the 6 o’clock position; (**J**) Enucleation of the right lobe; (**K**) Entering the bladder at the 2 o’clock position; (**L**) Enucleation of the right lobe from upside down, leaving a mushroom-like pedicle at the 6 o’clock position; (**M**) Enucleation of the apical part of the adenoma between the 10 and 2 o’clock position; (**N**) Resection of the avascularized adenoma at high speed; (**O**) Final view of the preserved mucosa of the sphincter; The arrows in the image indicate the preserved mucosa of the sphincter; (**P**) Resected tissue. A, adenoma; B, bladder; BN, bladder neck; C, capsule; V, verumontanum.

**Figure 2 medicina-61-01859-f002:**
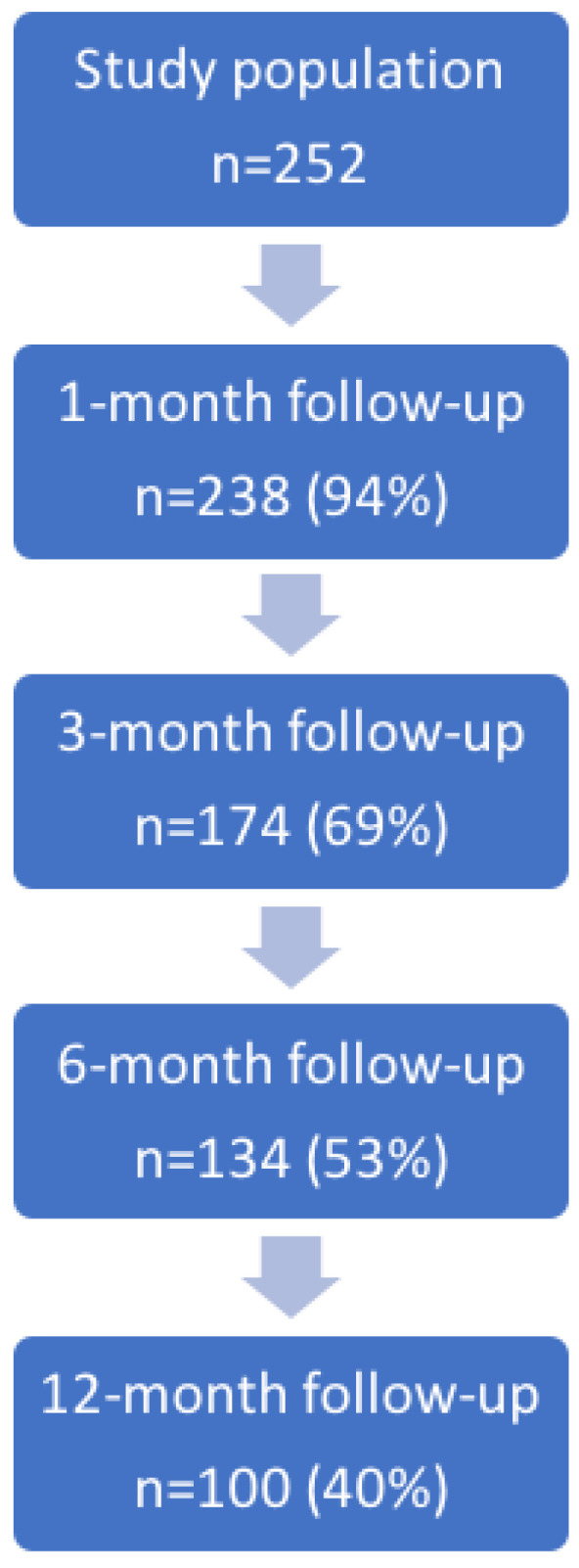
Flowchart of patient attendance at follow-up visits.

**Figure 3 medicina-61-01859-f003:**
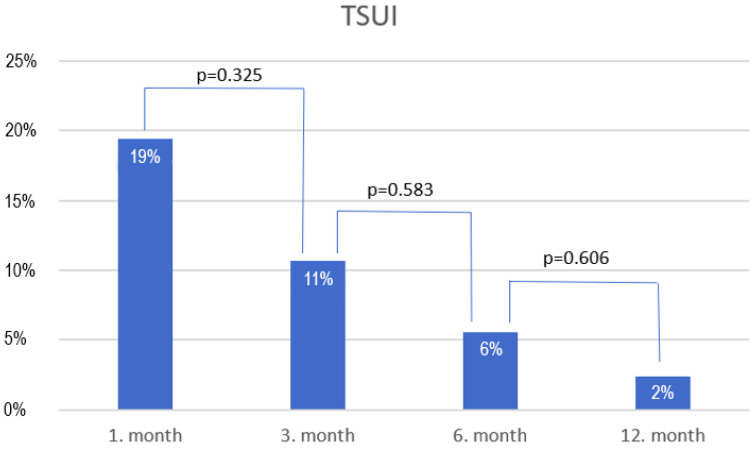
Bar chart showing the rate of transient stress urinary incontinence (TSUI) during the follow-up period.

**Table 1 medicina-61-01859-t001:** Baseline characteristics stratified by prostate volume subgroups.

Variables	Prostate Size: 30–60 mL	*p*-Value	Prostate Size: 60–100 mL	*p*-Value	Prostate Size: >100 mL
Age (year), median [IQR]	70 [64–75]	0.337	71 [65–76]	0.644	71 [66–76]
BMI, median [IQR]	27 [24–30]	0.882	27 [25–29]	0.007	30 [25–34]
ASA score, median [IQR]	2 [2–3]	0.721	2 [2–3]	0.237	3 [2–3]
Preoperative PSA (ng/mL), median [IQR]	2.37 [1.26–4.91]	<0.001	5.65 [3.53–8.9]	0.104	7.69 [4.05–12]
Preoperative IPSS, median [IQR]	21 [16–26]	0.478	19.5 [13–25]	0.494	21 [16–25]
Preoperative QoL, median [IQR]	6 [5–6]	0.14	6 [5–6]	0.465	6 [5–6]
Preoperative Qmax (mL/s), median [IQR]	7.8 [5.65–10.9]	0.26	9.02 [5.83–13.6]	0.869	9.05 [6.7–11.4]
Preoperative Qave (mL/s), median [IQR]	3.45 [2.46–5.35]	0.191	4.3 [3–5.88]	0.772	4.6 [3.1–5.7]
Preoperative PVR urine (mL), median [IQR]	112.5 [50–250]	0.147	150 [100–300]	0.047	101 [50–150]
Preoperative IIEF-5, median [IQR]	13 [7–17]	0.909	12 [6–19]	0.152	9 [5–17]

ASA, American Society of Anesthesiologists; BMI, body mass index; IIEF-5, International Index of Erectile Function Questionnaire; IPSS, International Prostate Symptom Score; IQR, Interquartile range; PSA, Prostate-specific antigen; PVR, Post-void residual; Qave, average flow rate; Qmax, maximum flow rate; QoL, Quality of Life.

**Table 2 medicina-61-01859-t002:** Baseline characteristics in patients with and without urinary retention.

Variables	Non-Retention	*p*-Value	Retention
Age (year), median [IQR]	68 [65–75]	0.04	71 [65–77]
BMI, median [IQR]	28 [26–31]	0.015	27 [24–29]
ASA score, median [IQR]	2 [2–3]	0.042	3 [2–3]
Preoperative PSA (ng/mL), median [IQR]	2.98 [1.47–5.69]	<0.001	5.5 [2.8–9.7]
Preoperative QoL, median [IQR]	5 [3–5]	<0.001	6 [6–6]

ASA, American Society of Anesthesiologists; BMI, body mass index; IQR, Interquartile range; PSA, Prostate-specific antigen; QoL, Quality of Life.

**Table 3 medicina-61-01859-t003:** Follow-up data.

Variables	Preoperative	Month (*n*)	Results	*p*-Value	Coeff. (95% CI)
PSA (ng/mL), median [IQR]	4.17 [1.79–7.81]	1 (238)	0.6 [0.27–1.13]	*p* < 0.001	−5.01 (−5.70, −4.32)
		3 (174)	0.47 [0.23–0.91]	*p* < 0.001	−5.36 (−6.12, −4.61)
		6 (134)	0.41 [0.22–0.87]	*p* < 0.001	−5.42 (−6.24, −4.60)
		12 (100)	0.34 [0.21–0.72]	*p* < 0.001	−5.40 (−6.32, −4.48)
PSA reduction from baseline (%)		1 (238)	60		
		3 (174)	77		
		6 (134)	77		
		12 (100)	77		
Qmax (mL/s), median [IQR]	8.4 [5.9–11.75]	1 (238)	19.5 [13.6–26.2]	*p* < 0.001	10.52 (8.81, 12.24)
		3 (174)	20.45 [16.15–27.59]	*p* < 0.001	12.61 (10.79, 14.43)
		6 (134)	21.2 [15.3–30.4]	*p* < 0.001	13.06 (11.16, 14.97)
		12 (100)	22.65 [14.7–29.5]	*p* < 0.001	13.19 (11.09, 15.29)
Qave (mL/s), median [IQR]	3.7 [2.8–5.7]	1 (238)	9.2 [6.3–12.9]	*p* < 0.001	5.36 (4.31, 6.42)
		3 (174)	10.6 [7.35–14.04]	*p* < 0.001	6.79 (5.67, 7.90)
		6 (134)	11 [7.7–14.3]	*p* < 0.001	7.66 (6.49, 8.83)
		12 (100)	11.75 [8.3–15.7]	*p* < 0.001	7.99 (6.70, 9.28)
PVR urine (mL), median [IQR]	120 [57.5–250]	1 (238)	25 [0–45]	*p* < 0.001	−127.81 (−140.59, −115.03)
		3 (174)	22.5 [0–50]	*p* < 0.001	−125.48 (−139.06, −111.91)
		6 (134)	30 [0–50]	*p* < 0.001	−125.30 (−139.67, −110.94)
		12 (100)	30 [0–50]	*p* < 0.001	−120.85 (−136.44, −105.27)
IPSS, median [IQR]	21 [16–25]	1 (238)	10 [7–15]	*p* < 0.001	−8.78 (−9.93, −7.63)
		3 (174)	7 [5–11]	*p* < 0.001	−11.09 (−12.32, −9.86)
		6 (134)	6 [4–10]	*p* < 0.001	−12.09 (−13.38, −10.80)
		12 (100)	6 [4–10]	*p* < 0.001	−11.73 (−13.14, −10.31)
QoL, median [IQR]	6 [5–6]	1 (238)	2 [1–4]	*p* < 0.001	−2.70 (−2.97, −2.44)
		3 (174)	1 [1–3]	*p* < 0.001	−3.10 (−3.40, −2.81)
		6 (134)	1 [0–2]	*p* < 0.001	−3.46 (−3.79, −3.14)
		12 (100)	1 [0–2]	*p* < 0.001	−3.40 (−3.75, −3.03)
IIEF-5, median [IQR]	12 [5–18]	1 (238)	11 [5–17]	*p* = 0.002	−1.11 (−1.79, −0.43)
		3 (174)	9 [5–16]	*p* = 0.006	−1.76 (−2.52, −1.00)
		6 (134)	10 [5–16]	*p* = 0.003	−1.60 (−2.43, −0.76)
		12 (100)	9 [5–16]	*p* < 0.001	−2.05 (−2.98, −1.12)

IIEF-5, International Index of Erectile Function Questionnaire; IPSS, International Prostate Symptom Score; IQR, Interquartile range; PSA, Prostate-specific antigen; PVR, Post-void residual; Qave, average flow rate; Qmax, maximum flow rate; QoL, Quality of Life.

**Table 4 medicina-61-01859-t004:** Descriptive subgroup outcomes for hemoglobin drop, catheter dwell time, 12-month IPSS, and Qmax.

Variables	Prostate Size: 30–60 mL	*p*-Value	Prostate Size: 60–100 mL	*p*-Value	Prostate Size: >100 mL
Hgb decrease (g/L), median [IQR]	−11 [−15–3]	0.001	−13 [−22–7]	0.007	−19 [−30–12]
Catheter dwell time (day), median [IQR]	3 [2–3]	0.148	3 [3–3]	0.877	3 [3–3]
IPSS at 12 month, median [IQR]	6 [3–9]	0.513	6 [4–10]	0.282	8 [4–14]
Qmax at 12 month (mL/s)	23.4 [13.3–30.2]	0.94	22.4 [15.7–29.7]	0.725	20.6 [13.9–29.5]

IPSS, International Prostate Symptom Score; Qmax, maximum flow rate.

## Data Availability

The data that support the findings of this study are available upon reasonable request.

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
