# Peer review of "En Bloc Bipolar Prostate Enucleation Using the Mushroom Technique with Early Apical Release: Short-Term Outcomes"

_medicina, 2025, doi:10.3390/medicina61101859_

Round 1
Reviewer 1 Report
Comments and Suggestions for Authors
I read with great interest the manuscript entitled “En-bloc Bipolar Prostate Enucleation Using the Mushroom Technique With Early Apical Release: Short-term Outcomes”. The authors present an innovative variation of anatomical endoscopic enucleation of the prostate (AEEP), combining en-bloc bipolar enucleation with the mushroom technique and early apical release. This represents a valuable contribution, as it addresses both clinical effectiveness and cost-efficiency—two key aspects in the current landscape of BPH surgery. One of the most compelling strengths of the study lies in its sizeable cohort (252 patients) and the systematic collection of perioperative, functional, and complication data. The results demonstrate significant improvements in all major functional outcomes (IPSS, QoL, Qmax, PVR), with durability at 12 months and a favorable complication profile. The authors also provide an interesting perspective on enucleation efficacy, showing that larger prostates are associated with higher efficiency—an observation that resonates with prior findings in AEEP literature. The technique itself deserves special attention. By avoiding morcellation and dedicated enucleation loops, the authors propose a pragmatic solution that can lower costs, reduce instrument dependence, and potentially broaden access in centers where laser platforms or morcellators are unavailable. Moreover, the focus on early apical release is clinically relevant, as it may mitigate transient stress urinary incontinence—a complication that remains a major concern in endoscopic enucleation procedures. In this context, it would be useful to also consider recent investigations exploring function-preserving approaches such as ejaculation-sparing Thulium laser enucleation, which highlight how refinements in surgical technique can balance symptomatic relief with preservation of quality-of-life domains that matter to patients (https://pubmed.ncbi.nlm.nih.gov/36362593/). Nevertheless, several aspects could be strengthened to enhance the manuscript’s impact. First, the retrospective and single-center design, although acknowledged, limits the generalizability of the findings. Explicitly discussing how the results compare with existing large multicenter studies on bipolar enucleation would place the work in a broader context. Second, while functional outcomes are well described, the manuscript could benefit from a more detailed exploration of sexual function results, especially given the decline in IIEF-5 scores observed during follow-up. This point is crucial, as erectile and ejaculatory outcomes are increasingly considered when counseling patients on BPH surgery. Another area for further elaboration concerns the management of complications. The authors report capsule perforations and postoperative hematuria at rates consistent with published series, but a more granular discussion of how these complications were handled intra- and post-operatively could provide useful practical insights for readers interested in adopting this approach. Additionally, the incidental prostate cancer detection rate (7.54%) is noteworthy and comparable to international data. However, discussing whether the mushroom technique allows adequate sampling of the transition zone compared to morcellator-based approaches would further reassure readers of its oncologic safety. Finally, while the study convincingly demonstrates short-term efficacy, the conclusions would be more robust if the authors outlined plans for prospective or multicenter studies with longer follow-up, to evaluate durability, learning curve, and long-term safety of this technique. In this regard, situating the present findings within the broader debate on minimally invasive surgical therapies (MISTs) for LUTS would enrich the translational message, as recent reviews have questioned whether these innovations represent a true paradigm shift or rather a selective solution for specific patient subgroups (https://pubmed.ncbi.nlm.nih.gov/37921510/). In conclusion, this manuscript offers an important and timely contribution to the field of surgical management of BPH. By proposing a cost-effective, reproducible, and safe alternative to conventional enucleation methods, the authors open a path toward wider adoption of enucleation in daily practice, even in settings where access to advanced instruments is limited.
Author Response
Reviewer 1
I read with great interest the manuscript entitled “En-bloc Bipolar Prostate Enucleation Using the Mushroom Technique With Early Apical Release: Short-term Outcomes”. The authors present an innovative variation of anatomical endoscopic enucleation of the prostate (AEEP), combining en-bloc bipolar enucleation with the mushroom technique and early apical release. This represents a valuable contribution, as it addresses both clinical effectiveness and cost-efficiency—two key aspects in the current landscape of BPH surgery. One of the most compelling strengths of the study lies in its sizeable cohort (252 patients) and the systematic collection of perioperative, functional, and complication data. The results demonstrate significant improvements in all major functional outcomes (IPSS, QoL, Qmax, PVR), with durability at 12 months and a favorable complication profile. The authors also provide an interesting perspective on enucleation efficacy, showing that larger prostates are associated with higher efficiency—an observation that resonates with prior findings in AEEP literature. The technique itself deserves special attention. By avoiding morcellation and dedicated enucleation loops, the authors propose a pragmatic solution that can lower costs, reduce instrument dependence, and potentially broaden access in centers where laser platforms or morcellators are unavailable.
Moreover, the focus on early apical release is clinically relevant, as it may mitigate transient stress urinary incontinence—a complication that remains a major concern in endoscopic enucleation procedures. In this context, it would be useful to also consider recent investigations exploring function-preserving approaches such as ejaculation-sparing Thulium laser enucleation, which highlight how refinements in surgical technique can balance symptomatic relief with preservation of quality-of-life domains that matter to patients (https://pubmed.ncbi.nlm.nih.gov/36362593/).
We thank the reviewer for the valuable remark and for highlighting the suggested article. In our study, we did not assess the incidence of retrograde ejaculation, which is indeed highly relevant, as it represents one of the most frequent sexual dysfunctions following surgery for BPH. Reduced ejaculate volume significantly decreases orgasmic intensity, impairs sexual satisfaction, and may lead to anxiety and depression, all of which can substantially affect patients’ quality of life. In everyday clinical practice, it is widely perceived that patients must choose between satisfactory voiding and preservation of ejaculation. Consequently, patients who prioritize maintaining antegrade ejaculation often decline surgical treatment, which may ultimately result in worsening lower urinary tract symptoms and progressive detrusor dysfunction. Retrograde ejaculation was traditionally attributed to impaired bladder neck closure; however, this notion is challenged by the observation that retrograde ejaculation is not typical after bladder neck incision (TUIP). Several studies have confirmed that the musculus ejaculatorius, rather than the bladder neck, plays a pivotal role in the process of ejaculation. The article suggested by the reviewer presents noteworthy results on ejaculation-sparing ThuLEP procedures. We fully intend to integrate this technique into our clinical practice to further improve patient satisfaction.
Nevertheless, several aspects could be strengthened to enhance the manuscript’s impact. First, the retrospective and single-center design, although acknowledged, limits the generalizability of the findings. Explicitly discussing how the results compare with existing large multicenter studies on bipolar enucleation would place the work in a broader context.
We thank the reviewer for this insightful comment. We agree that contextualizing our findings with multicenter data is important to strengthen the generalizability of our work. As suggested, we have added the corresponding information to the Discussion section. We would like to highlight the results of a recent multicenter study that directly compared HoLEP combined with morcellation and bipolar enucleation using the mushroom technique. This analysis confirmed that both procedures were safe and effective, with comparable functional improvements at 12 months. HoLEP was associated with shorter operative time and hospital stay, while IPSS, PSA, and PVR outcomes were similar between the groups. These findings support the external validity of our present results despite the single-center design of our study.
Lines: 478-484
Reference number: 49
Second, while functional outcomes are well described, the manuscript could benefit from a more detailed exploration of sexual function results, especially given the decline in IIEF-5 scores observed during follow-up. This point is crucial, as erectile and ejaculatory outcomes are increasingly considered when counseling patients on BPH surgery.
Thank you for the comment, this is an important point. We have revised the Discussion accordingly.
Erectile dysfunction (ED) is also a known complication after AEEP. According to the results of Enikeev et al., following Thulium Laser Enucleation of the Prostate (ThuLEP), erectile function remained stable in 56% of patients, deteriorated in 18%, and improved in 26% [34]. The decline in erectile function can be attributed to thermal damage to the prostatic capsule and the neurovascular bundles. Elshal et al. investigated the effect of low-energy (2 J/25 Hz) versus high-energy (2 J/50 Hz) HoLEP on sexual function, but found no significant differences, concluding that the applied energy does not substantially affect erectile function. Naturally, it should not be overlooked that after surgery, with the improvement of urinary symptoms, medications previously negatively affecting sexual function may be discontinued. A multicenter study including 235 HoLEP patients reported that increasing age was independently associated with a higher likelihood of erectile function decline at the 12-month follow-up (p = 0.03). This observation suggests that age-related vulnerability plays a more important role than the surgical technique itself in the onset of postoperative ED. Furthermore, patients with an ASA score above 2 were more prone to erectile deterioration one year after surgery, underlining the impact of preoperative physical condition on functional outcomes. At baseline, erectile function already indicated mild-to-moderate ED in our study population. At the 12-month follow-up, this significantly declined. Naturally, for the reasons detailed above, surgery may have contributed to the deterioration of erectile function; however, it should also be emphasized that our study involved an elderly patient cohort with multiple comorbidities, as 42.86% of the patients were classified as ASA grade 3. Therefore, providing appropriate and detailed patient counseling prior to surgery is of paramount importance.
Lines: 412-413
Reference numbers: 34-37
Another area for further elaboration concerns the management of complications. The authors report capsule perforations and postoperative hematuria at rates consistent with published series, but a more granular discussion of how these complications were handled intra- and post-operatively could provide useful practical insights for readers interested in adopting this approach.
Thank you for your question.
Capsule perforations occur more frequently in smaller prostates, as the capsule is typically less well defined and dissection of the plane is more challenging compared with larger glands. Liu et al. found that prostate volume was the strongest risk factor for an indistinct plane. According to their study, a prostate volume below 54 ml was associated with the highest likelihood of indistinct plane formation during HoLEP. In our series, all perforations occurred in prostates smaller than 60 ml. Enucleation was completed in all cases, and during the resection part a suction tube was connected to the outflow of the resectoscope to reduce intravesical pressure and thereby minimize the risk of hypervolemia.
Lines: 372-380
Reference numbers: 23-24
Intra- and postoperative bleeding may occur during AEEP; however, the extent of bleeding is generally lower compared with TURP or open prostatectomy, as the adenoma is precisely detached from the capsule while coagulation is performed simultaneously. Factors influencing the amount of blood loss include prolonged operative time, inadequate intraoperative coagulation, pre-existing coagulopathy, and capsule perforation. In severe intraoperative bleeding, the use of bipolar OvalButton electrode is a viable option. In our practice, postoperative hematuria is initially managed conservatively with intravenous tranexamic acid administration and prolonged bladder irrigation. If these measures prove insufficient, re-coagulation is required, which was necessary in 2.8% in our series.
Lines: 402-411
Reference numbers: 32-33
Additionally, the incidental prostate cancer detection rate (7.54%) is noteworthy and comparable to international data. However, discussing whether the mushroom technique allows adequate sampling of the transition zone compared to morcellator-based approaches would further reassure readers of its oncologic safety.
Thank you for this valuable comment; it is indeed an important point. Unfortunately, the available studies in the literature mainly focus on functional outcomes after enucleation and often do not report histopathological findings. We identified one study that compared the mushroom technique with morcellation; however, histological results were not provided in that analysis either (Weerasawin T. Comparison of the results of bipolar transurethral enucleation and resection of the prostate with and without morcellation in treatment of benign prostatic hyperplasia. Insight Urol. 2023;44: 7-13. doi: 10.52786/isu.a.66)
Finally, while the study convincingly demonstrates short-term efficacy, the conclusions would be more robust if the authors outlined plans for prospective or multicenter studies with longer follow-up, to evaluate durability, learning curve, and long-term safety of this technique.
We thank the reviewer for this valuable comment. The Conclusions section has been revised accordingly to emphasize the short-term nature of our findings and to highlight the need for longer follow-up and prospective or multicenter studies.
Lines: 473-474
In this regard, situating the present findings within the broader debate on minimally invasive surgical therapies (MISTs) for LUTS would enrich the translational message, as recent reviews have questioned whether these innovations represent a true paradigm shift or rather a selective solution for specific patient subgroups (https://pubmed.ncbi.nlm.nih.gov/37921510/).
Thank you for the comment.
Minimally invasive surgical therapies (MISTs) are gaining popularity; however, their role in the surgical management of BPH remains a topic of debate, particularly due to their higher costs and increased rates of reoperation compared with standard enucleation techniques. Although preservation of sexual function is a notable advantage, the primary objective remains the durable relief of lower urinary tract symptoms. Importantly, the substantially higher reoperation rates following MISTs not only increase the burden on healthcare systems but also subject patients to additional physical and psychological stress.
Lines: 433-441
Reference numbers: 38-39
In conclusion, this manuscript offers an important and timely contribution to the field of surgical management of BPH. By proposing a cost-effective, reproducible, and safe alternative to conventional enucleation methods, the authors open a path toward wider adoption of enucleation in daily practice, even in settings where access to advanced instruments is limited.
Reviewer 2 Report
Comments and Suggestions for Authors
I reviewed the manuscript “En-bloc Bipolar Prostate Enucleation Using the Mushroom Technique with Early Apical Release: Short-term Outcomes” by Kiss et al., the authors reports a single-center retrospective cohort (n=252) evaluating a cost-saving variant of AEEP: en-bloc bipolar enucleation performed mechanically with the resectoscope sheath, paired with the “mushroom” extraction and early apical release to protect sphincter mucosa. Clinical endpoints (IPSS, QoL, Qmax, Qave, PVR, PSA) improve significantly at twelve months; complication rates appear acceptable, and TSUI declines over time. I consider the topic relevant (resource-limited settings; training friendliness), and the surgical description is usable. However, several methodological and reporting issues limit robustness: novelty is over-claimed versus cited literature; statistics are minimal for a multi-timepoint dataset; sexual function outcomes are unfavorable but under-discussed; and important confounders (learning curve, antiplatelet therapy, retention status, prostate size) are not modeled. Some numerical/formatting errors need correction (e.g. ,a percentage that seems off by two orders of magnitude). Here my observations.
Major
The abstract and introduction frame the approach as unprecedented (“no studies combine en-bloc bipolar enucleation with the mushroom technique; no evaluation of mechanical enucleation without a dedicated loop”). Later, the Discussion cites multiple bipolar enucleation + mushroom-style series (albeit with channels and loop swaps), then differentiates your en-bloc + sheath-tip–only approach. Right now this reads as an over-general novelty claim. Please: re-write the novelty statement more precisely: the specific innovation is “en-bloc, sheath-tip mechanical dissection without dedicated enucleation loop+mushroom pedicle,” not the generic combination of bipolar enucleation and mushroom removal.
Add a concise scoping paragraph summarizing how your steps differ from refs 17–19 (channel creation, loop changes, irrigation dynamics, median lobe handling), and from classic HoLEP mushroom (energy source + morcellation). This avoids the impression (for me at least) of overstatement and clarifies clinical niche.
This is a retrospective, single-surgeon series (2018–2023), including men who “opted not to undergo medical therapy,” >50% in retention, and ~17% on antiplatelets. Such features strongly influence bleeding, catheterization, TSUI, and functional recovery. There is no contemporaneous control (e.g., TURP, standard bipolar AEEP with loop, or laser AEEP). To improve internal validity authors should provide case accrual figure with attrition and number at risk at each follow-up timepoint (1/3/6/12 months??).
Report case-mix by key strata (retention vs non-retention, prostate size bands, antiplatelet use, prior TURP) and show their baseline differences.
Present propensity-free descriptive subgroup outcomes and, ideally, a multivariable model (or at least stratified medians) for: early TSUI, hemoglobin drop, catheter days, and 12-month IPSS/Qmax. Learning-curve over calendar time (first 80 vs middle 80 vs last 80 cases) should also be analyzed—this is crucial in enucleation series.
Outcomes are repeatedly assessed (baseline, 1, 3, 6, 12 months) and tested pairwise with Wilcoxon. Multiple testing inflates α; the time structure is ignored. Please replace (or complement) serial Wilcoxon with a repeated-measures test (Friedman) or a linear mixed-effects model for each endpoint, reporting effect sizes and 95% CIs.
For TSUI, provide time-to-continence (Kaplan–Meier to zero-pad events; report median time, if estimable) and plot cumulative incidence with CIs rather than only a box plot.
For “enucleation efficacy” (g/min) and its apparent increase in larger prostates, please discuss denominator effects and adjust for case-mix (size, median lobe, retention). As is, the statement “larger prostates resulted in more effective enucleation” can be a mathematical artifact rather than true efficiency
Complications are given with Clavien–Dindo granularity. One number requires correction: 1 transfusion in 252 patients is 0.4%, not 0.004% as stated. Please fix the percentage and check all derived rates. Also, provide numerator/denominator and 95% CIs for each complication categoory.
Specify antibiotic prophylaxis, handling of pre-op bacteriuria (~50% positive cultures), and whether fever/epididymitis clustered in bacteriuric or catheter-prolonged patients.
IIEF-5 declines from median 12 pre-op to 9 at 12 months with significant p-values; however, the Discussion mostly centers on TSUI and does not interpret the erectile data (clinical relevance, MCID, age confounding, ASA burden, baseline ED distribution). This is important for patient counseling. Tehn authors should add a brief clinical significance paragraph (not only statistical), cite MCID thresholds, and discuss potential mechanisms (traction at apex, thermal spread, comorbidities).
A subgroup look (IIEF-5 ≥17 at baseline) to estimate risk of de-novo ED among those potent pre-operatively.
Some core definitions are missing or too loose:
TSUI, provide operational criteria(pad test? patient-reported?? any leakage?).
PSA reduction, define calculation and confirm assay uniformity.
Flow, clarify how uroflow and PVR were measured (device; ultrasound method), and whether observers were standardized..
transabdominal ellipsoid is acceptable, but please acknowledge its error vs TRUS; consider sensitivity analysis by size bands.
The stepwise figure is helpful, but reproducibility would benefit from energy settings (cut/coag wattage), irrigation pressure/height, and sheath model specifics already partially listed—make them complete and consolidated. Concrete guidance for identifying the plane using sheath-tip only (visual cues beyond “pathognomonic view”), and handling tricky scenarios (median lobe, bleeding at 5–7 o’clock). Even, a short video supplement (if Medicina journal allows) would materially increase educational value.
The discussion argues cost-effectiveness and training friendliness. These are plausible, but currently asserted, not shown. Pleasetemper language or provide basic cost comparison (capital + disposables) vs loop-based bipolar AEEP and vs laser AEEP; and
A clear limitations paragraph highlighting the absence of a comparator arm and the 12-month horizon (durability beyond 1 year remains unknown)
Minor
use periods for decimals in p-values, e.g., p=0.011 not p=0,011. Ensure consistent units (g/L vs g/dL) and round percentages appropriately.
add Ns at each timepoint to Table1; include 95% CIs. For the TSUI figure, show percent with event per visit (with bars) rather than only a box plot.
when citing 60–90% PSA reductions post-enucleation, link it to your data explicitly; consider a post-hoc ROC for PSA drop vs residual symptoms at 12 months (exploratory)
7.54% aligns with literature; still, specify Gleason grade group distribution and whether any case required subsequent therapy within 12 months.
minor edits will improve fluency ( “catheter dwell time” to “catheterization time” if journal style allows).
Comments on the Quality of English Languageminor edits will improve fluency ( “catheter dwell time” to “catheterization time” if journal style allows).
Author Response
Reviewer 2
I reviewed the manuscript “En-bloc Bipolar Prostate Enucleation Using the Mushroom Technique with Early Apical Release: Short-term Outcomes” by Kiss et al., the authors reports a single-center retrospective cohort (n=252) evaluating a cost-saving variant of AEEP: en-bloc bipolar enucleation performed mechanically with the resectoscope sheath, paired with the “mushroom” extraction and early apical release to protect sphincter mucosa. Clinical endpoints (IPSS, QoL, Qmax, Qave, PVR, PSA) improve significantly at twelve months; complication rates appear acceptable, and TSUI declines over time. I consider the topic relevant (resource-limited settings; training friendliness), and the surgical description is usable. However, several methodological and reporting issues limit robustness: novelty is over-claimed versus cited literature; statistics are minimal for a multi-timepoint dataset; sexual function outcomes are unfavorable but under-discussed; and important confounders (learning curve, antiplatelet therapy, retention status, prostate size) are not modeled. Some numerical/formatting errors need correction (e.g. ,a percentage that seems off by two orders of magnitude). Here my observations.
Major
The abstract and introduction frame the approach as unprecedented (“no studies combine en-bloc bipolar enucleation with the mushroom technique; no evaluation of mechanical enucleation without a dedicated loop”). Later, the Discussion cites multiple bipolar enucleation + mushroom-style series (albeit with channels and loop swaps), then differentiates your en-bloc + sheath-tip–only approach. Right now this reads as an over-general novelty claim. Please: re-write the novelty statement more precisely: the specific innovation is “en-bloc, sheath-tip mechanical dissection without dedicated enucleation loop+mushroom pedicle,” not the generic combination of bipolar enucleation and mushroom removal.
Thank you for this helpful comment, we acknowledge the reviewer’s observation. We have revised the Abstract and the Introduction to ensure that the novelty of our approach is framed more precisely. We hope that this rewording more accurately reflects the specific contribution of our study.
Lines: 12-15, 49-55
Add a concise scoping paragraph summarizing how your steps differ from refs 17–19 (channel creation, loop changes, irrigation dynamics, median lobe handling), and from classic HoLEP mushroom (energy source + morcellation). This avoids the impression (for me at least) of overstatement and clarifies clinical niche.
Thank you for this valuable comment. In our technique, the procedure is divided into two main parts: en-bloc enucleation followed by resection of the avascularized lobes. In the techniques described in the cited references, a channel is first resected at the 6 or 12 o’clock position (or both). Thereafter, the resection loop is exchanged for an enucleation loop. One lateral lobe is then enucleated, after which the loop is switched back to the resection loop to resect the tissue. The same sequence is repeated for the other lobe, resulting in a total of four loop exchanges during the procedure. By contrast, in our technique no loop exchange is required. In our technique, however, the median lobe is not resected at the beginning of the procedure; instead, it is enucleated en-bloc and subsequently resected together with the lateral lobes after com-pletion of the enucleation.
Lines: 345-348, 354-356
This is a retrospective, single-surgeon series (2018–2023), including men who “opted not to undergo medical therapy,” >50% in retention, and ~17% on antiplatelets. Such features strongly influence bleeding, catheterization, TSUI, and functional recovery. There is no contemporaneous control (e.g., TURP, standard bipolar AEEP with loop, or laser AEEP). To improve internal validity authors should provide case accrual figure with attrition and number at risk at each follow-up timepoint (1/3/6/12 months??).
Thank you for this valuable suggestion. We have prepared a flowchart (Figure 2) that illustrates patient accrual and attrition throughout the study period, including the number of patients available at each follow-up timepoint (1, 3, 6, and 12 months). We agree that this addition improves the clarity and internal validity of the study, and the flowchart has been incorporated into the revised manuscript.
Report case-mix by key strata (retention vs non-retention, prostate size bands, antiplatelet use, prior TURP) and show their baseline differences.
Thank you for this important comment. In line with your suggestion, we have reported the case-mix according to the key strata and presented their baseline differences. These data are now included in the revised manuscript as Table 1 and Table 2.
Present propensity-free descriptive subgroup outcomes and, ideally, a multivariable model (or at least stratified medians) for: early TSUI, hemoglobin drop, catheter days, and 12-month IPSS/Qmax. Learning-curve over calendar time (first 80 vs middle 80 vs last 80 cases) should also be analyzed—this is crucial in enucleation series.
Thank you for this suggestion. In accordance with your comment, we have added the descriptive subgroup outcomes. These results are now presented in the revised manuscript as Table 4.
Enucleation has a steep learning curve, with most studies reporting that surgical efficiency stabilizes after approximately 25–50 cases. To address your suggestion, we divided our cohort into three equal subgroups of 84 patients each (first 84, middle 84, last 84) and compared enucleation efficiency. The median values were 0.79 [0.60–1.02] g/min, 0.77 [0.62–0.94] g/min, and 0.68 [0.55–0.97] g/min, respectively, with no statistically significant differences. We believe this is largely explained by the fact that our initial 50 learning cases (performed in 2017) were not captured in this retrospective dataset, as systematic data collection began in 2018.
Outcomes are repeatedly assessed (baseline, 1, 3, 6, 12 months) and tested pairwise with Wilcoxon. Multiple testing inflates α; the time structure is ignored. Please replace (or complement) serial Wilcoxon with a repeated-measures test (Friedman) or a linear mixed-effects model for each endpoint, reporting effect sizes and 95% CIs.
Thank you for the suggestion. We have added linear mixed-effects model results with coefficients and 95% CIs (Table 3).
For TSUI, provide time-to-continence (Kaplan–Meier to zero-pad events; report median time, if estimable) and plot cumulative incidence with CIs rather than only a box plot.
Thank you for this valuable suggestion. We agree that a time-to-continence analysis with Kaplan–Meier curves would provide important additional information. However, as our dataset is retrospective, we do not have precise data on the exact time of continence recovery for each patient. Therefore, we illustrated the incidence of TSUI at each follow-up visit using a bar chart, which demonstrates the declining rates over time.
For “enucleation efficacy” (g/min) and its apparent increase in larger prostates, please discuss denominator effects and adjust for case-mix (size, median lobe, retention). As is, the statement “larger prostates resulted in more effective enucleation” can be a mathematical artifact rather than true efficiency
Thank you for this important remark. We agree that the apparent increase in enucleation efficacy (g/min) with larger prostates may partly reflect a denominator effect, since more tissue is removed without a proportionate increase in operative time. Thus, the observed association could be at least partly a mathematical artifact rather than a true improvement in surgical efficiency. In the revised Discussion, we have addressed this issue and acknowledged that case-mix factors such as prostate size, the presence of a median lobe, and urinary retention status should be taken into account when interpreting these findings.
Lines: 362-369
Complications are given with Clavien–Dindo granularity. One number requires correction: 1 transfusion in 252 patients is 0.4%, not 0.004% as stated. Please fix the percentage and check all derived rates. Also, provide numerator/denominator and 95% CIs for each complication categoory.
Thank you for pointing this out. We have corrected this value in the text and rechecked all complication rates, updating them where necessary. For transparency, we now provide, for each Clavien–Dindo complication category, the numerator/denominator alongside the percentage.
Specify antibiotic prophylaxis, handling of pre-op bacteriuria (~50% positive cultures), and whether fever/epididymitis clustered in bacteriuric or catheter-prolonged patients.
Thank you for raising this important point.
Intravenous antibiotics were administered to the patients 30 min before the start of surgery. Typically, a broad-spectrum third-generation cephalosporin, such as ceftriaxone (2 g), was used. In cases of positive preoperative urine culture, targeted parenteral an-tibiotic therapy (most commonly a cephalosporin) was initiated the day before surgery and continued during hospitalization.
Lines: 87-91
IIEF-5 declines from median 12 pre-op to 9 at 12 months with significant p-values; however, the Discussion mostly centers on TSUI and does not interpret the erectile data (clinical relevance, MCID, age confounding, ASA burden, baseline ED distribution). This is important for patient counseling. Tehn authors should add a brief clinical significance paragraph (not only statistical), cite MCID thresholds, and discuss potential mechanisms (traction at apex, thermal spread, comorbidities).
Thank you for your valuable suggestion. In line with your comment, we have extended the Discussion section by adding a paragraph that addresses this point in more detail, including additional references. We believe this strengthens the interpretation of our findings and provides a clearer context within the current literature.
Unfortunately, our literature search did not identify any published studies that established an MCID threshold specifically for the IIEF-5. We only found reports addressing MCID values for the IIEF-EF domain. Therefore, we did not incorporate MCID thresholds into the manuscript, although we acknowledge that this remains an important consideration.
Lines: 412-433
A subgroup look (IIEF-5 ≥17 at baseline) to estimate risk of de-novo ED among those potent pre-operatively.
Thank you for this important suggestion. In our cohort, 73 patients had a baseline IIEF-5 score ≥17 (21 [19–23]). At the 6-month follow-up, 41 of these patients were available for assessment, with a median IIEF-5 of 16 [8–22], indicating a decrease in erectile function among previously potent men. In line with another reviewer’s comment, we have expanded the Discussion to address the risk of de novo ED after enucleation and the clinical implications for patient counseling.
Lines: 261-264
Some core definitions are missing or too loose:
TSUI, provide operational criteria(pad test? patient-reported?? any leakage?).
Although there is no complete consensus among studies regarding the definition of urinary incontinence, most authors consider it clinically relevant in the context of enucleation if the patient requires the use of pads. In our study, we also defined TSUI as the need for pad use.
Line: 165
PSA reduction, define calculation and confirm assay uniformity.
The percentage reduction in PSA was calculated for each patient as (postoperative PSA – preoperative PSA)/preoperative PSA. These individual percentage changes were then averaged across the cohort.
All PSA measurements were performed in the same institutional laboratory using a standardized immunoassay platform, thereby ensuring assay uniformity throughout the study period.
Lines: 73-75
Flow, clarify how uroflow and PVR were measured (device; ultrasound method), and whether observers were standardized..
Uroflowmetry was performed using the Urodoc device (Urodoc, Budapest, Hungary), with one measurement obtained in each case. PVR urine was assessed by transabdominal ultrasound (Mindray Diagnostic Ultrasound System, Consona N9, Shenzhen, China) using the ellipsoid formula. All measurements were carried out by the same physician (Z. Kiss) to ensure standardization.
Lines: 79-83
transabdominal ellipsoid is acceptable, but please acknowledge its error vs TRUS; consider sensitivity analysis by size bands.
Thank you for this valuable and fully justified comment. We have supplemented the manuscript in the Limitations section accordingly.
Lines: 475-478
The stepwise figure is helpful, but reproducibility would benefit from energy settings (cut/coag wattage), irrigation pressure/height, and sheath model specifics already partially listed—make them complete and consolidated. Concrete guidance for identifying the plane using sheath-tip only (visual cues beyond “pathognomonic view”), and handling tricky scenarios (median lobe, bleeding at 5–7 o’clock). Even, a short video supplement (if Medicina journal allows) would materially increase educational value.
We have revised the Methods section according to your suggestion. The instruments have now been listed in detail, in a stepwise manner, including generator settings, sheath specifications, optic, electrode loop, and irrigation height.
We appreciate the reviewer’s request for more concrete guidance on surgical nuances. In response, we have expanded the Methods section to clarify the identification of the enucleation plane and the management of challenging intraoperative scenarios.
Lines: 93-102, 107-117, 118-125, 135-136
The discussion argues cost-effectiveness and training friendliness. These are plausible, but currently asserted, not shown. Pleasetemper language or provide basic cost comparison (capital + disposables) vs loop-based bipolar AEEP and vs laser AEEP; and
We fully acknowledge the reviewer’s concern regarding our original statement on cost-effectiveness. As no formal cost-analysis was performed, such a claim would indeed be speculative. We have therefore revised the manuscript to temper the language and have explicitly clarified that this consideration should be regarded as a hypothetical assumption.
Lines: 331, 334-336
A clear limitations paragraph highlighting the absence of a comparator arm and the 12-month horizon (durability beyond 1 year remains unknown)
According to your suggestion, we have expanded the Limitations section to include the absence of a comparator arm and the relatively short follow-up period limited to 12 months.
Lines: 473-474
Minor
use periods for decimals in p-values, e.g., p=0.011 not p=0,011. Ensure consistent units (g/L vs g/dL) and round percentages appropriately.
We thank the reviewer for this helpful comment. The p-values have been corrected to use periods consistently, the units have been standardized, and percentages have been rounded appropriately throughout the manuscript.
add Ns at each timepoint to Table1; include 95% CIs. For the TSUI figure, show percent with event per visit (with bars) rather than only a box plot.
We have revised the Table according to your suggestion.
We apologize for the confusion caused by the incorrect description. As requested, we did not use a box plot; the figure has always been a bar chart showing the percentage of patients with TSUI at each follow-up visit. In the bar chart, percentages were displayed after rounding.
when citing 60–90% PSA reductions post-enucleation, link it to your data explicitly; consider a post-hoc ROC for PSA drop vs residual symptoms at 12 months (exploratory)
Thank you for this suggestion. We have revised the manuscript by explicitly linking the literature-reported 60–90% PSA reduction range to our own findings (60% at 1 month and 77% at 3, 6, and 12 months). In addition, we performed the exploratory post-hoc ROC analyses for PSA drop versus residual symptoms (defined as IPSS ≥8) at all follow-up timepoints (1, 3, 6, and 12 months). However, no significant associations were observed, and therefore these analyses were not included in the revised version.
Lines: 360-361
7.54% aligns with literature; still, specify Gleason grade group distribution and whether any case required subsequent therapy within 12 months.
Thank you for the question. I have expanded the Results and Discussion sections of the manuscript to provide a more detailed description of the findings related to iPCa.
Lines: 240-244, 450-468
minor edits will improve fluency ( “catheter dwell time” to “catheterization time” if journal style allows).
We thank the reviewer for this helpful suggestion. The term “catheterization time” has been revised to “catheter dwell time” throughout the manuscript in accordance with the reviewer’s comment and to improve fluency.
Comments on the Quality of English Language
minor edits will improve fluency ( “catheter dwell time” to “catheterization time” if journal style allows).
We thank the reviewer for the helpful comment regarding the English language. The revised manuscript has been thoroughly reviewed by a professional editing service (Editage) to improve the overall quality, clarity, and fluency of the text.
Reviewer 3 Report
Comments and Suggestions for Authors
Endoscopic anatomic enucleation of the prostate has become an established alternative to TURP and open simple prostatectomy for benign prostatic hyperplasia (BPH). The technique aims to remove the adenoma along the true surgical capsule, improving functional durability while reducing bleeding and reoperation rates.
Bipolar energy (transurethral enucleation with bipolar energy, TUEB/BipolEP) is widely adopted in centers where laser platforms are unavailable, and has demonstrated comparable efficacy and safety to HoLEP in experienced hands.
The Mushroom technique was first described as a modification of endoscopic enucleation, creating a “mushroom-shaped” adenoma by partial resection/enucleation before final detachment.This approach facilitates en-bloc removal, reduces the need for prolonged morcellation, and provides enhanced hemostasis, particularly near the apical region. Technical mastery is required, and the maneuver is best suited for large prostates where tissue control is challenging.
Early apical release refers to the early dissection of the prostatic apex from the external sphincter at the beginning of the enucleation. Rationale: Minimizes traction on the external sphincter, reduces thermal injury, and provides better visualization of the apex. Expected benefit: Faster recovery of continence after catheter removal without compromising oncological or functional efficacy.
Authors demostrated that this technique is safe and efficacy, but the complication rate isn 't low. It would be interesting to know the later complications.
Without a doubt, are necessary randomized, multicenter trials
Author Response
Reviewer 3
Endoscopic anatomic enucleation of the prostate has become an established alternative to TURP and open simple prostatectomy for benign prostatic hyperplasia (BPH). The technique aims to remove the adenoma along the true surgical capsule, improving functional durability while reducing bleeding and reoperation rates.
Bipolar energy (transurethral enucleation with bipolar energy, TUEB/BipolEP) is widely adopted in centers where laser platforms are unavailable, and has demonstrated comparable efficacy and safety to HoLEP in experienced hands.
The Mushroom technique was first described as a modification of endoscopic enucleation, creating a “mushroom-shaped” adenoma by partial resection/enucleation before final detachment.This approach facilitates en-bloc removal, reduces the need for prolonged morcellation, and provides enhanced hemostasis, particularly near the apical region. Technical mastery is required, and the maneuver is best suited for large prostates where tissue control is challenging.
Early apical release refers to the early dissection of the prostatic apex from the external sphincter at the beginning of the enucleation. Rationale: Minimizes traction on the external sphincter, reduces thermal injury, and provides better visualization of the apex. Expected benefit: Faster recovery of continence after catheter removal without compromising oncological or functional efficacy.
Authors demostrated that this technique is safe and efficacy, but the complication rate isn 't low. It would be interesting to know the later complications.
Without a doubt, are necessary randomized, multicenter trials
We sincerely thank the reviewer for the supportive and positive feedback on our manuscript. We are grateful for the encouraging remarks. We fully agree with the reviewer that randomized, multicenter trials are needed to further validate these findings. In addition, we plan to extend our follow-up in future studies in order to better assess the incidence of late complications.
Reviewer 4 Report
Comments and Suggestions for Authors
This manuscript addresses an important and timely topic in endourology. The authors present a large single-center series (252 patients) evaluating en-bloc bipolar prostate enucleation performed with the mushroom technique and early apical release, without a morcellator or dedicated enucleation loop. The novelty lies in the combination of these elements—mechanical enucleation with the sheath tip, cost-effectiveness by avoiding morcellators, and the refinement of early apical release to minimize postoperative incontinence.
- The 7.5% incidental prostate cancer rate is an important finding. A brief comparison with TURP series and implications for preoperative assessment could be added. Important point
- The discussion compares results broadly with existing AEEP studies, but a more explicit positioning of this technique against standard bipolar enucleation and HoLEP would strengthen the novelty claim. For instance, direct comparison of enucleation efficacy and operative times with published bipolar series would be useful. Please cite in the discussion the following: doi: 10.1016/j.euf.2024.04.006.
- Since the technique relies on sheath-tip enucleation, it would be valuable to comment on reproducibility and expected learning curve, possibly with operative time trends over the study period.
- The study provides only short-term outcomes (12 months). The conclusions should be more cautious, underlining the need for longer follow-up on functional durability, stricture rates, and cancer detection.
- Erectile function data (IIEF-5) are briefly reported but show a decline. This aspect should be more thoroughly discussed, including whether the reduction is clinically meaningful and how it compares to other enucleation series.
- Some p-values are reported as “0.000” or with excess precision. Standard formatting (e.g., p<0.001) should be applied.
- Capsule perforation and stricture rates deserve a more detailed analysis, possibly stratified by prostate size or surgeon experience.
Author Response
This manuscript addresses an important and timely topic in endourology. The authors present a large single-center series (252 patients) evaluating en-bloc bipolar prostate enucleation performed with the mushroom technique and early apical release, without a morcellator or dedicated enucleation loop. The novelty lies in the combination of these elements—mechanical enucleation with the sheath tip, cost-effectiveness by avoiding morcellators, and the refinement of early apical release to minimize postoperative incontinence.
The 7.5% incidental prostate cancer rate is an important finding. A brief comparison with TURP series and implications for preoperative assessment could be added. Important point.
Thank you for your valuable comment. In our study, the incidence of iPCa was determined based on final histopathological findings, which were consistent with data reported in the literature. It is important to note that, similar to BPH, the occurrence of iPCa is primarily age-dependent. Although evidence indicates that the detection rate of iPCa has markedly decreased with the widespread use of PSA screening, iPCa diagnosis remains common in elderly patients undergoing surgery for BPH. Predictive factors for iPCa include advanced age, elevated preoperative PSA levels, smaller resected tissue volume, higher PSA density, and diabetes mellitus. Herlemann et al. found no significant difference in the incidence of iPCa between HoLEP and TURP. This can be explained by the fact that larger prostates are more likely to be diagnosed as BPH after the final histological analysis. It remains an interesting and clinically relevant question whether preoperative multiparametric magnetic resonance imaging (mpMRI) should be routinely performed in patients with elevated preoperative PSA levels, in order to improve the detection and risk stratification of iPCa. Porreca et al. investigated the relationship between preoperative negative mpMRI and the incidence of iPCa after HoLEP. They found that the incidence of iPCa was significantly lower in patients with a negative preoperative mpMRI compared to those without mpMRI (6.2% vs. 14.8%, p=0.03). Nevertheless, no significant differences were observed between the negative mpMRI and no-mpMRI groups with regard to the proportion of low-risk Gleason score 3+3=6 tumors (85.7% vs. 88.2%, p=0.86).
Lines: 450-468
Reference numbers: 44-48
The discussion compares results broadly with existing AEEP studies, but a more explicit positioning of this technique against standard bipolar enucleation and HoLEP would strengthen the novelty claim. For instance, direct comparison of enucleation efficacy and operative times with published bipolar series would be useful. Please cite in the discussion the following: doi: 10.1016/j.euf.2024.04.006.
Thank you for raising this important point. As suggested, we have expanded the Discussion part.
We would like to highlight the results of a recent multicenter study that directly compared HoLEP combined with morcellation and bipolar enucleation using the mushroom technique. This analysis confirmed that both procedures were safe and effective, with comparable functional improvements at 12 months. HoLEP was associated with shorter operative time and hospital stay, while IPSS, PSA, and PVR outcomes were similar between the groups. These findings support the external validity of our present results despite the single-center design of our study.
Lines: 478-483
Reference number: 49
We added a separate section in the Discussion focusing on minimally invasive surgical therapies (MISTs), citing the suggested article. MISTs are gaining popularity; however, their role in the surgical management of BPH remains a topic of debate, particularly due to their higher costs and increased rates of reoperation compared with standard enucleation techniques. Although preservation of sexual function is a notable advantage, the primary objective remains the durable relief of lower urinary tract symptoms. Importantly, the substantially higher reoperation rates following MISTs not only increase the burden on healthcare systems but also subject patients to additional physical and psychological stress.
Lines: 433-441
Since the technique relies on sheath-tip enucleation, it would be valuable to comment on reproducibility and expected learning curve, possibly with operative time trends over the study period.
Thank you for your comment. We agree that reproducibility and the learning curve are key considerations for sheath-tip enucleation. In our experience, once the correct enucleation plane is recognized, the sheath-tip technique can be performed safely and consistently without the need for dedicated enucleation loops. As this was a single-surgeon series, formal inter-operator reproducibility data are not available; however, we believe that the technique is readily adoptable for urologists already familiar with bipolar resection.
Enucleation has a steep learning curve, with most studies reporting that surgical efficiency stabilizes after approximately 25–50 cases. To address your suggestion, we divided our cohort into three equal subgroups of 84 patients each (first 84, middle 84, last 84) and compared enucleation efficiency. The median values were 0.79 [0.60–1.02] g/min, 0.77 [0.62–0.94] g/min, and 0.68 [0.55–0.97] g/min, respectively, with no statistically significant differences. We believe this is largely explained by the fact that our initial 50 learning cases (performed in 2017) were not captured in this retrospective dataset, as systematic data collection began in 2018.
The study provides only short-term outcomes (12 months). The conclusions should be more cautious, underlining the need for longer follow-up on functional durability, stricture rates, and cancer detection.
We thank the reviewer for this valuable comment. The Conclusions section has been revised accordingly to emphasize the short-term nature of our findings and to highlight the need for longer follow-up.
Lines: 496-501
Erectile function data (IIEF-5) are briefly reported but show a decline. This aspect should be more thoroughly discussed, including whether the reduction is clinically meaningful and how it compares to other enucleation series.
Thank you for the comment, this is an important point. We have revised the Discussion accordingly.
Erectile dysfunction (ED) is also a known complication after AEEP. According to the results of Enikeev et al., following Thulium Laser Enucleation of the Prostate (ThuLEP), erectile function remained stable in 56% of patients, deteriorated in 18%, and improved in 26% [34]. The decline in erectile function can be attributed to thermal damage to the prostatic capsule and the neurovascular bundles. Elshal et al. investigated the effect of low-energy (2 J/25 Hz) versus high-energy (2 J/50 Hz) HoLEP on sexual function, but found no significant differences, concluding that the applied energy does not substantially affect erectile function. Naturally, it should not be overlooked that after surgery, with the improvement of urinary symptoms, medications previously negatively affecting sexual function may be discontinued. A multicenter study including 235 HoLEP patients reported that increasing age was independently associated with a higher likelihood of erectile function decline at the 12-month follow-up (p = 0.03). This observation suggests that age-related vulnerability plays a more important role than the surgical technique itself in the onset of postoperative ED. Furthermore, patients with an ASA score above 2 were more prone to erectile deterioration one year after surgery, underlining the impact of preoperative physical condition on functional outcomes. At baseline, erectile function already indicated mild-to-moderate ED in our study population. At the 12-month follow-up, this significantly declined. Naturally, for the reasons detailed above, surgery may have contributed to the deterioration of erectile function; however, it should also be emphasized that our study involved an elderly patient cohort with multiple comorbidities, as 42.86% of the patients were classified as ASA grade 3. Therefore, providing appropriate and detailed patient counseling prior to surgery is of paramount importance.
Lines: 412-413
Reference numbers: 34-37
Some p-values are reported as “0.000” or with excess precision. Standard formatting (e.g., p<0.001) should be applied.
We thank the reviewer for this valuable comment. The p-values have been corrected and are now consistently reported in the standard format throughout the manuscript.
Capsule perforation and stricture rates deserve a more detailed analysis, possibly stratified by prostate size or surgeon experience.
Thank you for the valuable comment. Accordingly, we have expanded the Discussion section.
Lines: 372-401
Round 2
Reviewer 1 Report
Comments and Suggestions for Authors
Authors answered all comments and suggestions.
Reviewer 2 Report
Comments and Suggestions for Authors
The authors have adequately answered the questions asked, and I consider that they have made important changes improving their manuscript.
Reviewer 4 Report
Comments and Suggestions for Authors
The manuscript has improved significantly, addressing concerns with clarity.
Comments on the Quality of English Languagenone